# β-N-Methylamino-L-Alanine (BMAA) Modulates the Sympathetic Regulation and Homeostasis of Polyamines

**DOI:** 10.3390/toxins15020141

**Published:** 2023-02-09

**Authors:** Milena Shkodrova, Milena Mishonova, Mariela Chichova, Iliyana Sazdova, Bilyana Ilieva, Dilyana Doncheva-Stoimenova, Neli Raikova, Milena Keremidarska-Markova, Hristo Gagov

**Affiliations:** Department of Animal and Human Physiology, Faculty of Biology, Sofia University “St. Kliment Ohridski”, 8 Dragan Tzankov blvd., 1164 Sofia, Bulgaria

**Keywords:** ROS, heart, mitochondria, diamine oxidase, primary amines, polyamines, obestatin, catecholamines

## Abstract

The neurotoxin β-N-methylamino-L-alanine (BMAA) is a non-proteinogenic amino acid produced by cyanobacteria. Non-neuronal toxicity of BMAA is poorly studied with a reported increase in reactive oxygen species and a decrease in the antioxidant capacity of liver, kidney, and colorectal adenocarcinoma cells. The aim of this research is to study the toxicity of BMAA (0.1–1 mM) on mitochondria and submitochondrial particles with ATPase activity, on the semicarbazide-sensitive amino oxidases (SSAOs) activity of rat liver, and on an in vitro model containing functionally active excitable tissues—regularly contracting heart muscle preparation with a preserved autonomic innervation. For the first time the BMAA-dependent inhibition of SSAO activity, the elimination of the positive inotropic effect of adrenergic innervation, and the direct and reversible inhibition of adrenaline signaling in ventricular myocytes with 1 mM BMAA were observed. Additionally, it is confirmed that 1 mM BMAA can activate mitochondrial ATPase indirectly. It is concluded that a higher dose of BMAA may influence multiple physiological and pathological processes as it slows down the degradation of biogenic amines, downregulates the sympathetic neuromediation, and embarrasses the cell signaling of adrenergic receptors.

## 1. Introduction

The neurotoxin β-N-methylamino-L-alanine (BMAA) is a non-proteinogenic amino acid produced by peculiar photoautotrophic prokaryotes that photosynthesize with the production of oxygen and utilize atmospheric nitrogen (Cyanobacteria/Cyanoprokaryota) [1,2]. The ability of morphologically and ecologically different species worldwide from possibly all cyanobacterial orders to synthesize this compound suggests that BMAA could be transferred much more often to humans than it is accepted [1,2,3,4]. Common BMAA-producing cyanobacteria are from the orders *Chroococcales*, *Oscillatoriales*, and *Nostocales*, and the genera *Anabaena*, *Leptolyngbya*, Pseudoanabaena, and Phormidium. The total amount of BMAA increases from 0.3 μg/g in cyanobacteria, to 1160 μg/g in cycad seeds, and further to 3560 μg/g in the Micronesian flying fox *Pteropus mariannus*, which is considered a delicacy food by the locals [5]. It is considered that diatom algae, dinoflagellates and even plants can also synthesize and accumulate BMAA [2,6,7,8]. Therefore, the accumulation of toxic doses of BMAA in humans from aquatic [9,10] and other food sources is quite probable, at least for some areas and seasons.

BMAA is known as a neurotoxin that is harmful for a wide variety of neurons and glial cells [11,12,13]. Neurodegenerative conditions, such as Parkinson’s disease, Alzheimer’s disease, and amyotrophic lateral sclerosis are associated with BMAA incidences [14,15,16,17]. BMAA (627 μg/g) was found in the brain tissues of patients who had died from amyotrophic lateral sclerosis/Parkinson’s disease [18,19]. BMAA can directly trigger the process of neurodegeneration because it penetrates through the blood–brain barrier [20]. Once having reached the cerebrospinal fluid, BMAA binds to excitatory ionotropic (NMDA and AMPA types) and metabotropic glutamate receptors, which ultimately leads to an increase in the neuronal Na^+^ and Ca^2+^ concentrations and decreases the K^+^ in the postsynaptic cells [15,21]. Thus, higher doses of BMAA can induce an intracellular Ca^2+^ overload in neurons.

The mitochondria accumulate free Ca^2+^ from the cytosol during an agonist-induced or voltage-gated Ca^2+^ influx [22]. This physiological process activates the mitochondrial respiratory rate and ATP production to provide the enhanced need of energy provoked by excitatory signals [22]. The increased free Ca^2+^ concentration in the mitochondrial matrix activates enzymes within the tricarboxylic acid cycle, such as isocitrate dehydrogenase, alpha-ketoglutarate dehydrogenase, and pyruvate dehydrogenase, which further enhances ATP production by F_1_F_0_-ATP synthase (ATPase) [23]. However, abnormally elevated free Ca^2+^ can lead to mitochondrial dysfunction and cellular damage due to an overproduction of the reactive oxygen species (ROS) and cytochrome c release in cytosol, which trigger neuronal apoptosis [15,24]. Therefore, mitochondrial Ca^2+^ overload can lead to oxidative stress and protein misfolding followed by proteostatic collapse and neurodegeneration, which is often manifested in Alzheimer’s and Parkinson’s disease [17,25,26]. A Ca^2+^ overload, high values of ROS, and decreased cell viability in the presence of BMAA are also observed in non-neuronal cells, and these effects depend on the cyanotoxin concentration [12,13]. The BMAA-induced activation of respiratory chains in the inner mitochondrial membrane that enhances transmembrane H^+^ transport is paradoxically followed by a decrease instead of an increase in its negative membrane potential (∆Ψm) in the presence of toxins [23,27]. Therefore, the intimate mechanism of the BMAA action on mitochondrial function is still not completely clear. Additionally, BMAA has other molecular targets and uses multiple mechanisms to promote neurodegenerative disorders [28].

Data on non-neuronal BMAA toxicity are scare [1,12,13]. Liu et al. [29] reported that BMAA inhibited cysteine/glutamate antiporter, which led to depletion of the antioxidant glutathione (GSH). Higher concentrations (1 and 2 mM) of BMAA suppress catalase activity in a human colorectal epithelial adenocarcinoma cell line (Caco-2 cells) [30]. These effects suggest a more general BMAA toxicity in metabolically active tissues as it can, additionally to the increase in mitochondrial ROS production, support the oxidative stress by reducing the antioxidant cell capacity. Thus, BMAA application leads to oxidative stress in the liver, and to a lesser extent, in the kidney, as estimated by the change in activities of catalase and glutathione peroxidase, by GSH levels and lipid peroxidation [31]. Many other proteins and functions could also be affected by a direct interaction with the toxin, because BMAA can bind to hydroxyl groups of amino acid residues tyrosine, serine, and threonine of cellular proteins, which changes their conformation and regulation by protein kinases and function [32].

Semicarbazide-sensitive amine oxidases (SSAOs) are enzymes that catalyze the oxidative deamination of primary amines [33] and diamines, including polyamines [34,35]. The homeostasis of primary amines is important for human health and their improper metabolism is a significant factor for obesity [36], liver fibrosis [37], inflammation, and many other diseases [17,38]. The polyamines putrescine, spermidine, and spermine are ubiquitous regulators of cell growth and development via the processes of gene expression, cell signaling, and post-translational modifications [38]. An important organ for the turnover of primary amines is the liver [31,34]. Thus, alteration of the activity of the liver SSAO in the presence of BMAA could reveal changes in the degradation process of functionally important biogenic amines, therefore enlightening new targets and mechanisms of BMAA toxicity.

The aim of this research is to study the toxicity of BMAA on mitochondria and submitochondrial particles (SMPs) with ATPase activity, on rat liver fraction, containing active SSAO, and on an in vitro model of functionally active excitable tissues—a regularly contracted frog heart muscle preparation with preserved autonomic innervation.

## 2. Results

### 2.1. Effect of BMAA on ATPase Activity of Intact Mitochondria

BMAA was initially tested on intact mitochondria for the possible uncoupling action of the toxin. Our experiments demonstrated that the ATPase activity remained low during the 600 s registration under control conditions (Figure 1). The addition of the uncoupler 2.4-dinitrophenol (DNP) to a final concentration of 50 µM strongly stimulated the ATP hydrolysis, which indicated a normal functional state of the mitochondria and a low permeability of their inner membranes. The ATPase activity of both intact (Figure 1A) and DNP-uncoupled mitochondria (Figure 1B) was not affected by BMAA when applied at a concentration of 1 mM. This suggests that the toxin does not possess a quick uncoupling effect on intact liver mitochondria, and it is not able to pass readily through the inner mitochondrial membrane.

### 2.2. Effect of BMAA on ATPase Activity of Freeze–Thawed Mitochondria and Submitochondrial Particles (SMPs)

Freezing/thawing disrupts the mitochondrial inner membrane leading to an uncoupling and stimulation of the initial ATPase activity. After this treatment, the mitochondria preserve their structure and the membrane spanning F_0_ sector of the ATPase.

BMAA at a concentration of 1 mM significantly increased the ATPase activity of the freeze–thawed mitochondria by 121.09 ± 5.22% (*p* = 0.002; n = 6; and Figure 2). The ATPase activity values were calculated as percentages of the activity (expressed as µM Pi/mg protein/min) measured under control conditions (BMAA-free assay medium).

Based on these data, it was suggested that the observed effect of BMAA was due to a direct interaction with the catalytic complex F_1_ or with the membrane sector F_0_. To test this hypothesis, the toxin was applied to SMPs. SMPs are closed, inverted vesicles derived from the inner mitochondrial membrane. The complexes F_1_ are located outside of the membrane and are oriented towards the solution, which allows a direct interaction with BMAA. Considering this, the BMAA was applied initially in a ten-fold lower concentration of 0.1 mM. The BMAA in both tested concentrations (0.1 mM and 1 mM) did not significantly affect the ATPase activity of the SMPs (98.62 ± 1.52% in the presence of 0.1 mM BMAA vs. control; *p* = 0.391 and 95.73 ± 1.96% in the presence of 1 mM BMAA vs. control; *p* = 0.061 and; n = 5).

### 2.3. Effect of BMAA on SSAO Activity

The effect of three different concentrations of BMAA (0.1 mM, 0.3 mM, and 1 mM) was studied in vitro on the SSAO activity of the supernatant of the rat liver homogenate. In the presence of 1 mM BMAA, the activity of the SSAO was significantly decreased by 45.0 ± 10.5% of the control value (*p* = 0.002, n = 6; Figure 3). Lower concentrations of BMAA did not affect the SSAO activity: 0.1 mM toxin decreased it by −2% (*p* = 0.416, n = 5) and 0.3 mM activated it by 1% (*p* = 0.39, n = 6).

### 2.4. Effect of BMAA on Frog Heart Preparations In Vitro

The effect of three different concentrations of BMAA (0.1 mM, 0.3 mM, and 1 mM) was studied in vitro on a maximal force of contraction of isolated frog heart preparations. Obestatin-stimulated (1 nM and 100 nM) adrenaline secretion from the sympathetic nerves in the heart wall and exogenous administered 50 µM of adrenaline increased the force of heart contractions. In the presence of 0.1 mM and 0.3 mM BMAA, they retained their statistically significant positive inotropic effect when compared to the control group treated with the Ringer solution (Figure 4A).

In the presence of 1 mM BMAA, the application of obestatin and adrenaline do not increase the force of the frog heart contractions. Their amplitudes remain commensurable with the time control (1 nM obestatin *p* = 0.206; 100 nM obestatin *p* = 0.361; and adrenaline *p* = 0.169, n = 6, Figure 4B).

In the other set of experiments, the double administration of adrenaline and 1 mM of BMAA statistically significantly increased the force of the frog heart contractions only in the second minute of their first introduction, while the second application left them unchanged. At the end of the experiment, adrenaline was applied without the toxin, which also statistically significantly increased the force of the heart contractions only in the second minute of its introduction (Figure 4C).

## 3. Discussion

### 3.1. Effect of BMAA on Mitochondrial ATPase

Our preliminary results have shown that the ATPase activity of freeze–thawed mitochondria was stimulated in the presence of BMAA. This result raised the assumption for a direct interaction of BMAA with ATPases on the inner mitochondrial membrane, which leads to enhanced ATP synthesis. The absence of a direct stimulation of BMAA on the ATPase complexes of the submitochondrial particles however, rejects the hypothesis of a rapid and direct action of this cyanotoxin on the catalytic F_1_ complex or the membrane F_0_ sector of the enzyme. Therefore, the observed BMAA-induced ATPase activation of freeze–thawed mitochondria seems to be indirect, and most probably depends on the enhanced, free Ca^2+^ and the activation of some key enzymes of the tricarboxylic acid cycle in the mitochondrial matrix [17,23] increase the transmembrane proton gradient. The suggested mechanism for the indirect activation of mitochondrial ATPase by BMAA is supported by the observed decrease in the negative membrane potential (∆Ψm) of the inner membrane in the presence of BMAA, which is evidence for a stronger utilization of the proton gradient by the ATPase enzyme [27].

### 3.2. BMAA Decreases Liver SSAO Activity

Experiments with a direct application of BMAA on SSAO isolated from rat liver show a significant inhibition of the amine oxidase activity in 1 mM BMAA-containing media. It is very probable that the cell surface form of SSAO, the vascular adhesion protein 1 (VAP-1), which is a protein highly expressed in vasculature especially on the surface of endothelial and smooth muscle cells [39], will be more sensitive to BMAA in comparison to intracellular diamine oxidases, because for VAP-1 the cell membrane is not a barrier for the penetration of the toxin. The enzymatic activity of VAP-1 plays an important role in the adhesion and exudation of leukocytes from or in blood vessels [40]. Additionally, the soluble form of VAP-1 (sVAP-1), which is released in blood plasma by proteolytic cleavage of VAP-1 [41], can also be influenced. Therefore, a moderate decrease in catabolism of biogenic amines and thus their enhanced quantity could be a consequence of poisoning with a higher dose of BMAA. However, the most remarkable and direct mechanism of BMAA toxicity on non-neuronal cells, such as hepatocytes, seems to depend on the oxidative stress that this toxin induces metabolic active tissues [31].

### 3.3. BMAA Toxicity on Sympathetic Neuromediation and Adrenergic Signaling

The excised frog heart muscle preparation is a useful model for physiological, pharmacological, and toxicological studies of excitable tissues because it has functional muscle cells and preserved autonomic nerve projections [42]. A set of experiments was conducted with an application of BMAA under control conditions and together with obestatin. In the presence of BMAA, the hormone obestatin was added at low and high concentrations to reveal the activity of sympathetic innervation, as the presence of obestatin leads to the release of adrenaline that we measure indirectly by its positive inotropic effect on the heart ventricle [42,43]. Adrenaline was applied at the end of the experiments to prove the activity of adrenergic signaling in the heart muscle. The obestatin-induced adrenaline secretion and the adrenergic signaling in the heart muscle remained unaffected by lower concentrations of BMAA (0.1 mM and 0.3 mM). However, 1 mM BMAA inhibited both obestatin-induced and adrenaline-dependent increases of heart muscle contractions, but left the amplitudes of the steady-state contractile activity unchanged. The latter result suggests that (i) in the presence of the highest used concentration of BMAA, the neurons do not secrete adrenaline, and/or (ii) the adrenaline signaling in the heart muscle is blocked directly, because the adrenaline application at the end of the experiments did not show any effect. To clarify the direct influence of BMAA on muscle tissue, the adrenaline was added three times at 15 min intervals, first at the 25th min after the first toxin application (Figure 4C). The first two of them contained 1 mM BMAA but the last application did not contain the toxin. Under these conditions, a significant but shorter positive inotropic effect was observed after the first and the third adrenaline application. The partial recovery of the adrenaline influence on the amplitudes of ventricular contractions after two BMAA and the adrenaline applications and its absence after the same treatment with a combination of obestatin and BMAA (Figure 4B) suggest a competitive action of BMAA on adrenergic receptors. This difference could be explained with a lower binding affinity of BMAA to the adrenergic receptors, and for this reason, it is easy remove it in the presence of adrenaline. It is concluded that higher concentrations of BMAA directly block both autonomic neurotransmission and frog heart adrenergic signaling. The latter effect is slower and shows a washout-like recovery of adrenaline regulation. It was lately reported that BMAA in submicromolar concentrations inhibited the vesicular monoamine transporter (VMAT) at porcine-dense granules [44] with a similar potency as that observed in our paper on reserpine [42]. This observation suggests a second mechanism of BMAA action on the neurotransmission of catecholamines and serotonin additionally to its oxidative stress-dependent toxicity. Therefore, it can be expected that the complex, BMAA-dependent inhibition of biogenic amine regulations was due to the combination of a direct decrease in their secretion from the autonomic neurons and adrenal medulla cells, and suppressed adrenergic signaling in the target cells.

In summary, BMAA could influence both neuronal and muscle tissues. The autonomic neurons are sensitive to its presence. BMAA could damage sympathetic regulation by inducing oxidative stress in adrenergic nerve endings, such as neurotoxin 6-hydroxydopamine [42], and by inhibiting the loading of secretory vesicles similarly to reserpine [44]. On the other hand, BMAA is not only accumulated and stored in muscle tissues, as observed in fishes and animals (chicken) feed with blue mussels-based feed [45,46], but it also suppresses the muscle regulation of adrenaline.

## 4. Conclusions

The common results of this study are as follows: (i) amino oxidases (SSAOs) are reported as BMAA-sensitive enzymes for the first time, which suggests an increased regulatory effect of polyamines on the physiological processes, due to their slower degradation in the presence of the toxin; (ii) BMAA could inhibit the adrenergic regulation of sympathetic autonomic innervation; (iii) BMAA could downregulate adrenergic receptors signaling in muscle tissue; and (iv) it presents evidence for the indirect activation of mitochondrial ATPase by BMAA.

## 5. Materials and Methods

### 5.1. Isolation of Intact Rat Liver Mitochondria and SMPs

All experimental procedures were conducted in accordance with EU regulations in the Directive 2010/63/EU of the European Parliament and of the council of 22 September 2010 on the protection of animals used for scientific purposes. We have ethical approval from the national Bulgarian Food Safety Agency, Ministry of Agricultural, Food and Forestry (No. 224 from 23 January 2019) and from the local Ethics Committee of Faculty of Biology, Sofia University “St. Kliment Ohridski” (Protocol No. 2/2022). All efforts were made to minimize the number of animals used and their suffering.

The livers for assay of SSAO activity, intact liver mitochondria, and SMPs were isolated from mature male Wistar rats (50–60 days old, supplied by Vivarium with Physiological Laboratory, Project: Creation and Development of Centers of Competence: BG05M2OP001-1.002-0012-C01, “Sustainable utilization of bio-resources and waste of medicinal and aromatic plants for innovative bioactive products”). The animals were housed in cages at 20–24 °C for 12 h light/dark cycle. They received a standard laboratory diet (commercial rat chow, HL-TopMix Ltd., Kaloyanovo, Bulgaria) and water ad libitum and were fasted prior to use. Rats were anesthetized and then decapitated, and the liver was surgically removed.

### 5.2. Assay of Mitochondrial ATPase Activity

ATPase activity was determined by measuring the inorganic phosphate (Pi) released from ATP as stated in [47]. The reaction with freeze–thawed mitochondria was carried out in 1 mL assay medium consisting of 200 mM sucrose, 10 mM KCl, 50 mM Tris-HCl, and 100 µM EDTA-KOH (pH = 7.5). The assay medium for SMPs had the same composition, except that 2 mM MgCl_2_ was added since these preparations lose bound Mg^2+^ during isolation. Aliquots of BMAA stock solution were added to reach final concentrations of 0.1 and 1 mM, respectively. After pre-incubating the mitochondria in the assay medium for 10 min at 37 °C, the reaction was initiated by the addition of ATP in final concentration of 1 mM, continued for 5 min at 37 °C, and terminated by adding 0.4 mL of 3 M perchloric acid.

The ATPase reaction with intact mitochondria was carried out at room temperature, with continuous stirring in 4 mL assay medium consisting of 200 mM sucrose, 10 mM KCl, 50 mM Tris-HCl, 100 mM ЕDТА-KOH, and 1 mM АТP (pH = 7.5), and 50 μM DNP was included whenever indicated. Aliquots of BMAA stock solution were added to reach final concentrations of 1 mM. The reaction was started by adding the mitochondrial suspension. Samples of 500 μL were taken after 30, 60, 120, 180, 300, and 600 s of incubation and were added to 200 μL of 3 M perchloric acid for termination of the reaction.

In all experiments, the protein precipitates were removed by centrifugation at 8800× *g* for 10 min. The concentration of Pi in the supernatant was determined spectrophotometrically (spectrophotometer S-22UV/Vis, Boeco, Hamburg, Germany). Blanks in which the reaction was blocked by addition of perchloric acid before ATP addition were carried out in parallel, in order to determine the background Pi amount as a result of non-enzymatic hydrolysis. The activity of mitochondrial ATPase was expressed as μM Pi/mg protein/min or μM Pi/mg protein for freeze–thawed SMPs, and intact mitochondria, respectively.

### 5.3. Assay of Rat Liver SSAO Activity

Rats for in vivo experiments were used after a 25-day treatment and were compared to untreated control animals of the same age. The liver was quickly isolated, washed with cold sodium phosphate buffer 0.01 M, pH = 7.0, weighed, and homogenized in a ratio of 1:4 (w:v) with the same buffer. The homogenate was heated to 60 °C in a water bath for 10 min and centrifuged at 20,000× *g* for 20 min. The supernatant was collected for analysis. SSAO activity was determined spectrophotometrically using putrescine as a substrate [48]. Hydrogen peroxide, formed in the amine oxidase reaction, formed a color complex with phenol and 4-aminoantipirine in the presence of peroxidase. The extinction was measured spectrophotometrically at λ = 500 nm. Protein content was determined by the method of Lowry [49], using bovine serum albumin as a standard.

The standard reaction mixture for the SSAO activity assay (3.0 mL final volume), before the photometric measurement, contained 0.1 M sodium phosphate buffer (pH 7.4), 0.82 mM 4-aminoamtipirine, 10.6 mM phenol, 4 IU/mL of horseradish peroxidase, 2.5 mM putrescine, 1.0 mM semicarbazide, and 300 µL of supernatant.

The blanks, containing a buffer, an enzyme source, and peroxidase, were preincubated with semicarbazide at 37 °C for 20 min. The control samples contained the same components, except for semicarbazide. The blanks and the samples also contained BMAA (1 mM). After preincubation, putrescine, 4-aminoantipirin, and phenol were added, and all tubes were incubated at 37 °C for 60 min. The reaction was stopped by cooling the tubes on ice and semicarbazide was added to the samples.

### 5.4. Study of Excised Frog Heart Contraction

*Pelophylax ridibundus* frog hearts were isolated and cannulated, and the data were recorded and analyzed as described earlier [43]. These excised heart preparations have functionally active sympathetic nerve projections [42]. This allows for the testing of the effect of BMAA on two excitable structures simultaneously—the heart muscle and the adrenergic axons.

All experiments were performed at room temperature (20–22 °C). In the control group (n = 6), cardiac activity was measured and Ringer solution (200 μL) in the cannula was replaced every 15 min. Excised frog heart preparations developed regular contractions with stable patterns and force. Under our experimental conditions, the spontaneous contractions slightly declined during the experiment (16% on average) after the 10th min of the experiment. According to the Frank–Starling mechanism, each application of a fresh solution causes a short-term increase in heart contractions, which was always observed, regardless of the solution’s composition. In the second experimental group, after 15 min of adaptation, we applied BMAA at concentrations of 0.1 mM, 0.3 mM, or 1.0 mM and obestatin (1 and 100 nM) in the presence of BMAA.

### 5.5. Statistical Analysis

Data are presented as mean ± standard error (SEM), mostly from six independent experiments. The difference between the treated samples and the untreated controls was tested with a one-way analysis of variance (ANOVA) assay for the mitochondrial ATPase activity and independent samples Student’s *t*-test for the SSAO activity and heart preparations. Values of *p* < 0.05 were considered significant. All variables are normally distributed, which was checked and confirmed by the Shapiro–Wilk test in R package (ver. 4.2.2).

## Figures and Tables

**Figure 1 toxins-15-00141-f001:**
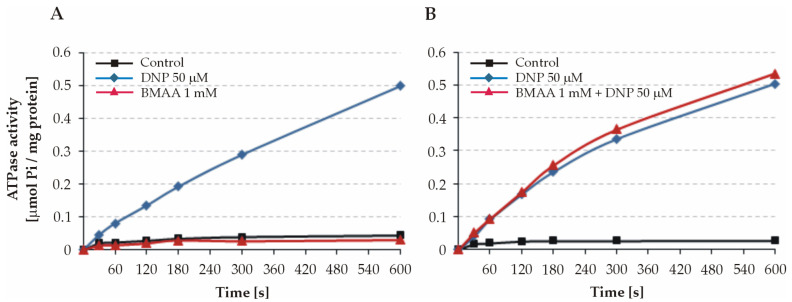
Effects of BMAA on the ATPase activity of intact mitochondria (**A**) and DNP-uncoupled mitochondria (**B**). The reactions were started by adding 80 µL of mitochondrial suspension (protein 7.0 mg/sample (**A**) and 6.35 mg/sample (**B**), respectively) and carried out for 10 min. Data present curves from two representative experiments.

**Figure 2 toxins-15-00141-f002:**
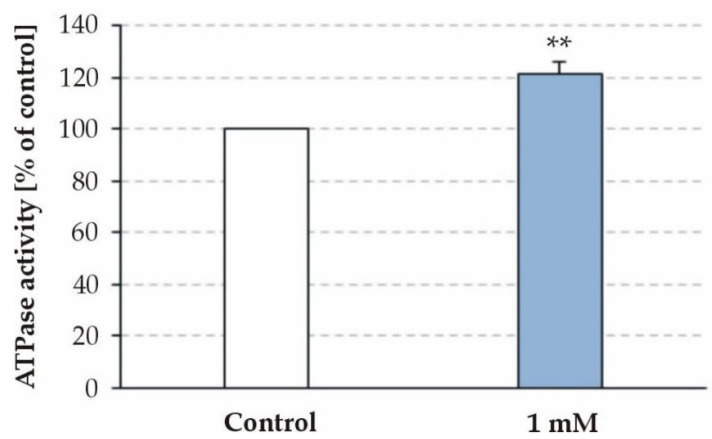
Effect of BMAA on ATPase activity of freeze–thawed mitochondria. ATPase activity values were calculated as percentages of the activity measured under control conditions (BMAA-free assay media). Data are plotted as mean ± SEM of six freeze–thawed mitochondria measurements and five independent experiments with SMPs (three parallel samples per group per experiment). Asterisks indicate significant differences (** *p* < 0.01) from the controls.

**Figure 3 toxins-15-00141-f003:**
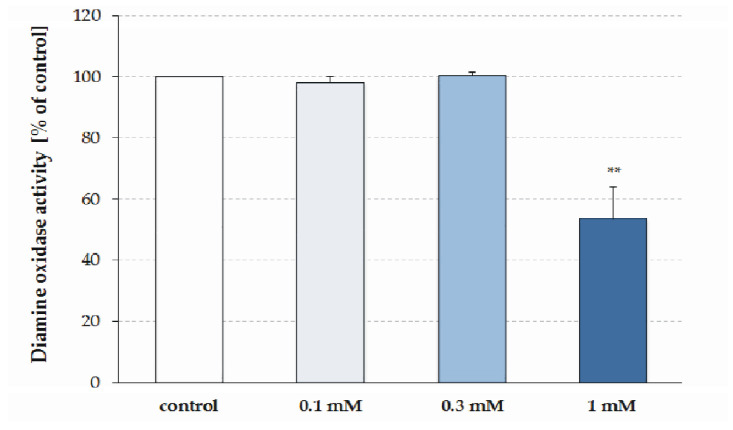
Effect of 0.1 mM, 0.3 mM, and 1 mM BMAA on SSAO activity of rat liver homogenate. SSAO activity values were calculated as percentage of the enzyme activity of the control. Data are presented as mean ± SEM. Asterisks indicate significant differences (** *p* < 0.01) from the controls.

**Figure 4 toxins-15-00141-f004:**
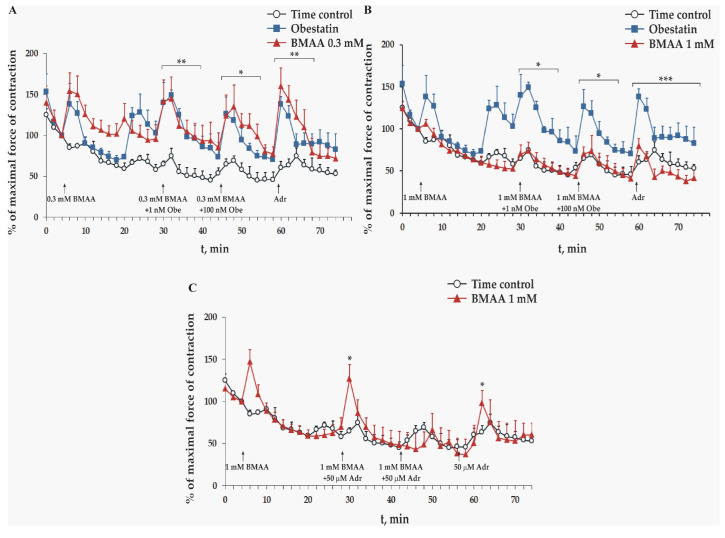
Effect of 0.3 mM (**A**) and 1 mM (**B**) BMAA on excised frog heart. Maximal force of heart contraction at control conditions (empty circles) and in presence of obestatin (filled squares) and are shown for comparison (**A**,**B**). Data are plotted as mean ± SEM of six 0.3 mM BMAA- and 1 mM BMAA-treated preparations. Obe is obestatin and Adr is adrenaline. Asterisks indicate significant differences (* *p* < 0.05, ** *p* < 0.01, and *** *p* < 0.001) from the controls on (**A**,**C**), and * *p* < 0.05 from obestatin on (**B**).

## Data Availability

Not applicable.

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
