# Peer review of "β-N-Methylamino-L-Alanine (BMAA) Modulates the Sympathetic Regulation and Homeostasis of Polyamines"

_toxins, 2023, doi:10.3390/toxins15020141_

Round 1

Reviewer 1 Report

The manuscript is well presented and adds to the understanding of BMAA toxicity.
However, I have the following comments:

1. How do the tested doses relate to environmental concentrations of BMAA? How does it relate to potential human exposures?
2. Title is completely non-specific and not accurate. Please specify what this new data refer to.
3. BMAA is produced by selected cyanobacteria; please specify.
4. Specify the tested concentrations in the Abstract.
6. Was the data normally distributed? This should be clarified in the description of the statistical methods. If not, the non-parametric KW ANOVA should be used.

Author Response

Answers to Reviewer's comments

Reviewer 1: The manuscript is well presented and adds to the understanding of BMAA toxicity. However, I have the following comments:

1. How do the tested doses relate to environmental concentrations of BMAA? How does it relate to potential human exposures?

            Long lasting exposure to lower doses of BMAA from food sources leads to accumulation of BMAA. In our study, BMAA treatments lasted between several minutes for experiments with mitochondria and SSAO to about 1 hour for heart preparations, i.e. they were limited in time. For this reason higher concentrations of BMAA were tested routinely in this and other toxicological studies. Thus, in a recent Review of Cyanotoxicity Studies Based on Cell Cultures (Sazdova et al., 2022, J Toxicol. 2022:5647178. doi: 10.1155/2022/5647178) the concentrations of BMAA used in cell cultures studies vary between 0.1 and 3 mM. They were 1 mM in papers cited there under numbers [21,22], 0.1-3 mM in [24], 1-3 mM in [26], 0.1-1 mM in [25] and 0.3 mM in [22] and all these applications lasted 24-96 hours, i.e. were longer when compare to our study. Also, similar concentrations were tested on other model systems. Therefore, the used concentrations of BMAA (0.1-1 mM) in this study are adequate.

            Concerning human exposures – it can be suggested that BMAA accumulation is needed to manifest harmful effects on functions that starts with the mentioned in Introduction neurodegenerative conditions. The included reference of Lance et al., 2018 [9]  in R1 of our manuscript “highlights that the continuous presence of total BMAA in marine mollusks, with the free forms being more sporadic, suggests a potential risk for consumers” and “monitoring programs over several seasons and countries, as well as some case-controlled epidemiological studies in areas of high incidence of ALS combined with nutrition surveys” for human exposure are incomplete.

         Environmental concentrations of BMAA and their relation to potential human exposures are explained with the sentences “The total amount of BMAA increases from 0.3 μg/g in cyanobacteria, to 1160 μg/g in cycad seeds and further to 3560 μg/g in Micronesian flying fox Pteropus mariannus, which is used as delicacy food by the locals [5].” and “Therefore, the accumulation of toxic doses of BMAA in humans from aquatic [9,10] and other food sources is quite probable at least for some areas and seasons.” added into the first paragraph of 1.Introduction, the latter at the end of this paragraph.

2. Title is completely non-specific and not accurate. Please specify what this new data refer to.

          We rename our paper. The suggested new title is: β-N-Methylamino-L-alanine (BMAA) modulates the sympathetic regulation and homeostasis of polyamines

3. BMAA is produced by selected cyanobacteria; please specify.

            The text “Common BMAA producing cyanobacteria are from orders Chroococcales, Oscillatoriales, Nostocales, and genera Anabaena, Leptolyngbya, Pseudoanabaena and Phormidium [5 and references herein].” was added into first paragraph of 1.Introduction.

4. Specify the tested concentrations in the Abstract.

       The tested concentrations of BMAA are specified in the Abstract

5. Was the data normally distributed? This should be clarified in the description of the statistical methods. If not, the non-parametric KW ANOVA should be used.

          All variables are normally distributed, which was checked and confirmed by the Shapiro-Wilk test in R package (ver. 4.2.2).

Reviewer 2 Report

New Data on β-N-Methylamino-L-alanine (BMAA) Toxicity

Comments to authors:

General Comments:

The authors study the acute toxicity of BMAA on mitochondria and submitochondrial particles (SMP) with ATPase activity on rat liver fraction, containing active SSAO, and on an in vitro model of functionally active excitable tissues - a regularly contracted frog heart muscle preparation with preserved autonomic innervation. This work is fully integrated within the up to date and relevant understanding in healthcare. However, I would suggest performing modifications in order to improve the clarity of the study.

Specific Comments:

-Please, have a think about the title and whether it will be attractive to a wide audience - emphasise the fundamental science aspects of the work.

-Make sure the abstract starts with the main important findings of the study in order to increase the readers' interest.

-Introduction could be increased and restructured to better establish the main objectives of the study. Please ensure the hypotheses are clearly given and testable and that they relate to the aims and the objectives of the work.

-Results and Discussion sections should be better explored, increased and restructured to point out the main data. Many of the figures and tables do not add very much to the text, they could be deleted.

-Make sure the conclusions section are supported by the data presented.

-Please, make sure the references are up to date and that you have checked recent issues of Toxins.

In conclusion, I hope the comments and suggestions above may be of helping to the authors for improving a version of the manuscript.

Author Response

Answers to Reviewer's comments

Reviewer 2

General Comments:

The authors study the acute toxicity of BMAA on mitochondria and submitochondrial particles (SMP) with ATPase activity on rat liver fraction, containing active SSAO, and on an in vitro model of functionally active excitable tissues - a regularly contracted frog heart muscle preparation with preserved autonomic innervation. This work is fully integrated within the up to date and relevant understanding in healthcare. However, I would suggest performing modifications in order to improve the clarity of the study.

 Specific Comments:

-Please, have a think about the title and whether it will be attractive to a wide audience - emphasise the fundamental science aspects of the work.

We rename our paper. The suggested new title is: β-N-Methylamino-L-alanine (BMAA) modulates the sympathetic regulation and homeostasis of polyamines

-Make sure the abstract starts with the main important findings of the study in order to increase the readers' interest.

            Changes were made in abstract to emphasize the main findings of our study.

-Introduction could be increased and restructured to better establish the main objectives of the study. Please ensure the hypotheses are clearly given and testable and that they relate to the aims and the objectives of the work.

            In Introduction section were added data for cyanobacteria taxa producing BMAA and for human exposure. Also some minor changes were made in text for a better reading.

-Results and Discussion sections should be better explored, increased and restructured to point out the main data. Many of the figures and tables do not add very much to the text, they could be deleted.

            In Results – Figures 2B and 4A were deleted, as it was suggested. The text was changed accordingly. Common subjects in 4. Discussion were titled and numbered. The text iof the last paragraph of 3.3. BMAA toxicity on sympathetic neuromediation and adrenergic signaling, starting with “In summary…”, was shortened and simplified.

-Make sure the conclusions section are supported by the data presented.

         Conclusions were clarified and simplified. The order of conclusions was changed, starting with the most important findings. The confirming conclusion remained at the end of the paragraph.

-Please, make sure the references are up to date and that you have checked recent issues of Toxins.

           The references are up to date. The newest is from January 2023 – ref. Kennedy et al., 2023 [44]. Citations of some recent and appropriate BMAA papers from Toxins were included in the revised version of our research paper. These new references are with numbers [4], [7], [8], [9] and [10].

Round 2

Reviewer 2 Report

Β-N-METHYLAMINO-L-ALANINE (BMAA) MODULATES THE SYMPATHETIC REGULATION AND HOMEOSTASIS OF POLYAMINES

General Comments to authors:

The authors study the acute toxicity of BMAA on mitochondria and submitochondrial particles (SMP) with ATPase activity on rat liver fraction, containing active SSAO, and on an in vitro model of functionally active excitable tissues - a regularly contracted frog heart muscle preparation with preserved autonomic innervation. They have made the major revisions I requested to the previous version of the manuscript, and this has greatly improved with respect to the original submission. Consequently, I suggest accepting the manuscript in its present form.Β-N-METHYLAMINO-L-ALANINE (BMAA) MODULATES THE SYMPATHETIC REGULATION AND HOMEOSTASIS OF POLYAMINES

General Comments to authors:

The authors study the acute toxicity of BMAA on mitochondria and submitochondrial particles (SMP) with ATPase activity on rat liver fraction, containing active SSAO, and on an in vitro model of functionally active excitable tissues - a regularly contracted frog heart muscle preparation with preserved autonomic innervation. They have made the major revisions I requested to the previous version of the manuscript, and this has greatly improved with respect to the original submission. Consequently, I suggest accepting the manuscript in its present form.